# A simulation chamber for absorption spectroscopy in planetary atmospheres.

Marcel Snels[1], Stefania Stefani[2], Angelo Boccaccini[2], David Biondi[2], and Giuseppe Piccioni[2]

[1]National Research Council of Italy, Institute of Atmopsheric Sciences and Climate, ISAC-CNR, Via del Fosso del Cavaliere 100, 00133 Rome

[2]Istituto di Astrofisica e Planetologia Spaziali (IAPS)- Istituto Nazionale di AstroFisica (INAF) , Via del Fosso del Cavaliere 100, 00133 Rome

**Correspondence:** Marcel Snels (m.snels@isac.cnr.it)

**Abstract.**

A novel simulation chamber PASS$x$S (Planetary Atmosphere Simulation System for Spectroscopy) has been developed for absorption measurements performed with a Fourier Transform Spectrometer (FTS) and, possibly, a cavity ring down (CRD) spectrometer, with a sample temperature ranging from 100 K up to 550 K, while the pressure of the gas can be varied from 10 mbar up to 60 bar. These temperature and pressure ranges cover a significant part of the planetary atmospheres in the solar system and the absorption chamber can thus be used to simulate planetary atmospheres of solar planets and extra solar planets with similar physical conditions. The optical absorption path for the FTS absorption measurements is 3.2 m, due to the implementation of a multipass setup inside the chamber. The FTS measurements cover a wide spectral range, from the visible to the mid-infrared with a sensitivity sufficient for medium strength absorption bands. The FTS has been used previously to measure high pressure atmospheres, including collision induced absorption bands and continuum absorption at ambient temperatures. PASS$x$S allows to measure the temperature dependence of collision induced bands and continuum absorption, which is important both for the modelling of planetary atmospheres as well as for fundamental processes involving collisions between molecules and atoms.

## 1 Introduction

The study of planetary atmospheres is important for exploring atmospheric composition and chemical and dynamical processes for solar and extra solar planets, and might provide also valid information on the habitability of a planet or satellite. Many space missions targeting planets of the solar system have been equipped with optical sensors in various spectral ranges with the goal to measure the composition and the physical conditions of the atmosphere. These can also provide information about geological processes, such as vulcanic and tectonic activities, temperature profiles, the presence of liquid water, necessary for the existence of life, and the formation of clouds. Most of the sensors used on orbiting platforms are passive sensors, integrating the emitted radiation from the surface or from the lower atmosphere up to the spacecraft in orbit. Careful analysis of these observations, along with a good knowledge of spectral information, line-broadening processes as well as atmospheric chemistry of all chemical compounds present in different layers of the atmosphere (atoms, molecules, clusters, radicals, ions) may

provide profiles of the concentrations of all absorbing, and sometimes non-assorbing, chemical species. Much of the required knowledge is available from laboratory experiments as well as satellite observations of the Earth, supported by a well developed theory of absorption and emission processes. Unfortunately the laboratory absorption measurements at high pressures for a large temperature range are far from complete. This puts a severe limit to the exploitation of high pressure planetary atmospheres at extreme cold or hot conditions with respect to Earth. Previously we have performed FTS measurements at high pressure (up to 40 bar) and high temperature (up to about 500 K) (Stefani et al., 2013, 2018) by using a commercial absorption cell characterized by a short optical path (about 2 cm). Here we propose a newly designed absorption chamber which can be cooled down to 100 K and heated up to 550 K, with a long effective absorption pathlength, in contiguous measurement sessions, without changing the gas sample.

The main requirements for this absorption chamber can be summarized as follows.

1. The optical materials as well as vacuum sealings for optical windows have to be compatible with cryogenic temperatures as well as temperatures up to 550 K, and should support pressures up to at least 60 bar.

2. An efficient and well-distributed heating and cooling system is necessary to obtain a uniform temperature of the sample gas.

3. Safety measures are required to deal with accidental events during the operation of the system.

4. A stable optical interface with the FTS (and in the future with a CRD) is needed, as well as the maintenance of a stable alignment during heating and cooling.

Here we present a facility for performing absorption measurements in gases or mixtures of gases at intermediate pressures (up to 60 bar) at temperatures ranging from 100 to 550 K. Although several facilities are mentioned in literature for optical measurements in gases, up to our knowledge, none of these cover the combination of pressures and temperatures obtained with PASS$x$S.

## 2  Heatable and coolable absorption cells in literature

Several heatable absorption cells have been developed in the past, with the goal to perform spectroscopic studies of hot gases. More than fifty years ago Abu-Romia and Tien (1966) reported a high temperature absorption cell, using a 20 cm cell with sapphire windows heated in a furnace capable to reach temperatures of 1200 K and pressures up to 4 atm. Hartmann and Perrin (1989) used a 11.4 cm long heated cell to measure the absorption of 1% $CH_4$ in $N_2$ at a wavelength of 3.384 $\mu$m at pressures between 0.5 and 1.5 atm in a temperature range of 290-800 K. Hartmann et al. (1993) studied the continuum absorption of water vapor for temperatures and pressures in the 500-900 K and 0-70 atm ranges, respectively, in the 1900-2600 and 3900-4600 cm$^{-1}$ spectral ranges.

Rieker et al. (2007) measured near-infrared absorption spectra of water vapor, by using tunable diode lasers and a heatable Inconel cell with a pathlength of 37.6 cm, equipped with tapered sapphire windows and copper seals. Schwarm et al. (2019)

constructed an absorption cell for spectroscopic measurements in the mid-infrared range, capable to withstand pressures up to 200 atm and temperatures up to 1200 K. Such high temperatures are of particular interest for combustion processes. They used $CaF_2$ or sapphire rods with a length of 15 cm, sealed with a high-temperature epoxy to cooled end caps, while only the central part of the absorption cell was heated in a furnace. They reported a temperature uniformity of $\pm$ 7 K at 542 K in the 9 cm long absorption cell . A similar cell has been reported by Almodovar et al. (2019) with a longer pathlength (21 cm) and a slightly worse, but still very good temperature uniformity ( $\pm$ 19 K at 802 K). Also Christiansen et al. (2016) used a short pathlength cell (3 cm) equipped with sapphire windows inserted in a furnace to measure CO and $CO_2$ at high pressures and temperatures. They reported a temperature uniformity of $\pm$ 1 K at 1000 K.

The short pathlength cell (2 cm) (AABSPEC, model 2T-AWT) used by Stefani et al. (2013) supports pressures up to 200 bar. This cell uses a double chamber design; the windows of the sample chamber can be heated up to 650 K, but are not subjected to a pressure difference. The outer chamber is filled with a non-absorbing buffer gas and kept at ambient temperature with a closed circuit water cooling. The external windows are thus subject to a pressure difference, but are at ambient temperature. Kalrez O-rings were used to seal the hot windows.

All the cells mentioned before are heatable cells. Coolable cells have been used for many decades, with pioneering work by McKellar and co-workers (McKellar et al., 1970). Typically these cells where coupled to a Fourier Transform Spectrometer and used to measure spectra of molecular complexes (clusters). Ballard et al. (1994) reported such a cell with the possibility to study gases at pressures up to 5 bar. Several coolable cells with multiple reflection optics have been reported with very long pathlengths ( see e.g. Horn and Pimentel (1971) (2540 m), Kim et al. (1978) (1500 m) and Briesmeister et al. (1978) (500 m)). A review of coolable cells can be found in Le Doucen et al. (1985). Mondelain et al. (2007) and Guinet et al. (2010) reported a multipass Herriot cell with a pathlength of 12.49 m designed for temperatures between 20 and 296 K. They performed line broadening experiments on carbon dioxide and methane at low temperatures.

Very few absorption cells have been constructed with the possibility of both heating and cooling; one example can be found in (Schermaul et al., 1996). The cell developed by Schermaul et al. (1996) consists of a 3 m Pyrex glass tube embedded in a quartz-sand bath which can be heated by resistive heating elements and cooled by circulating a coollant. The cell temperatures vary from 123 to 423 K, at atmospheric pressures. The windows are kept at room temperature. Shetter et al. (1987) constructed a long path cell for absorption measurements at temperatures from 215 to 470 K at pressures up to 10 atm. They designed a cell consisting of three concentric stainless steel tubes, the innermost housing White-type optics. They flowed a cooling/heating fluid between the two inner tubes and used the outer tube to create a vacuum to provide thermal insulation. The maximum optical pathlength obtained was 96 m. Table 1 shows a non exhaustive list of heatable and coolable absorption cells reported in literature, for spectroscopic measurements of pure gases and mixtures, with some relevant parameters and is meant to show the state of the art in this field.

PASS$x$S has been designed as a large volume (about 11 L), long pathlength absorption cell with the possibility to cool and heat the cell while at pressures up to 60 bar. The large volume of the cell makes it less attractive for using expensive gases or isotopic species. While most previous heatable absorption cells are short pathlength cells ( less than 0.5 m) the maximum pathlength of the White cell optics inside PASS$x$S is 9.6 m. Moreover, PASS$x$S can also be cooled to 100 K, without changing

optics or vacuum seals. The chamber allows to vary the temperature from 100 K to 550 K while keeping the same gaseous sample. By performing these temperature scans for different pressures (densities), the temperature dependence of spectral features due to collisional broadening, such as line broadening, collision induced bands and the so-called continuum absorption can thus be determined. It is well known that these processes are proportional to the square of the density, but their temperature dependence is still under study, both from an experimental and theoretical point of view.

## 3  General aspects of the simulation chamber

The simulation facility consists of two concentric chambers (see Figure 1), the inner one containing the gas or mixture of gases and the external one ensuring the thermal insulation from the laboratory environment. To this purpose, a medium vacuum ($10^{-3}$- $10^{-2}$ mbar) is maintained between the two chambers by means of a rotary pump. Moreover the sample chamber is mounted on three thermally insulating supports, made principally of MACOR rods (see Figure 1), while all connections for cooling, heating and the gas supply to the sample chamber have been also thermally insulated from the outer chamber by using double wall stainless steel tubing. The inner walls are in contact with the liquid nitrogen and are welded on one side to a larger tube which provides the vacuum, thus avoiding exposure of the cold tube to the environment. The volume of the sample chamber is about 11 L, while the outer chamber has a volume of about 62 L and may thus function as a buffer in case of a gas leak from the sample chamber.

The outer vessel is supplied with several vacuum ports for pumping, electrical connections, gas input and output and liquid nitrogen input and output, as can be seen in Figures 1 and 2. The pressure of the gas is measured with a pressure gauge (JUMO MIDAS Type 401001) with a range from 0 to 60 bar. This gauge has a zero offset of less than 0.3% full scale and less than 0.5 % full scale linearity error.

The sub-systems of the simulation facility have been schematically depicted in Figure 2. The four sub systems are :

1. A cooling system (indicated in blue).

2. A heating system (indicated in red).

3. A set of 6 type T thermocouples to measure the temperature (indicated in green).

4. A gas inlet and outlet system (indicated in violet).

The cooling is achieved with a flow of liquid nitrogen regulated by a dedicated PID controller. The liquid nitrogen flows through stainless steel tubings in close contact with the inner cylinder and through double wall stainless steel tubing on the flanges, as shown in Figure 1. The tubings have been dimensioned to obtain a heat exchange proportional to the masses to be cooled. The total volume of the cooling circuit is 4.3 L and the flow of nitrogen is 5-10 g/s. The cooling circuit consists of two parts, one is cooling the removable flange, which provides optical access, the other circuit cools the body of the cell, with the counter flange and the welded flange. Temperature regulation is achieved by controlling the flow of liquid and gaseous nitrogen in the cooling circuit.

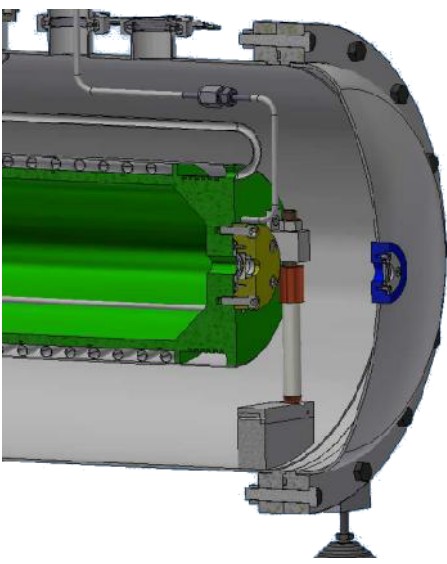

**Figure 1.** Section of the atmospheric simulation chamber.The sample chamber is indicated in green, while the external chamber is indicated in gray. One of the three insulated supports of the sample chamber can also be observed as well as the cooling circuit.

The heating is done by four tubular resistors with a total heating power of 2600 W. Two resistors of 400 W are used to heat the counter flange (heater 2, see figure 2) and the bottom flange (heater 4), both welded to the cylinder and two resistors of 900 W are used to heat the cylinder (heater 3) and the removable flange (heater 1). The temperature is measured on six different places, four type T thermocouples have been mounted on several parts of the cell and two inside the cell, measuring the temperature of the gas in two different positions. The accuracy of the type T thermocouples is $\pm 1°C$ or $\pm 0.75\%$, whichever is greater. The gas outlet is supplied with a safety valve, to avoid that the pressure in the sample chamber exceeds a preset value (70 bar).

## 4  Optical Windows and Vacuum Sealings

As shown in Figure 3, the simulation chamber is equipped with two optical windows for the entrance and exit of the light beam from the FTIR spectrometer and other two windows for the entrance and exit of the laser beam for future cavity ring down experiments.

These optical windows have to satisfy several requirements. First of all their thermal properties have to be compatible with the temperatures of the absorption chamber (100 to 550 K) and they have to be resistant to thermal shock. Secondly, their mechanical properties have to guarantee that they are stable under high pressure. The usual vacuum seals for optical windows, such as neoprene, Viton, Teflon and others are not suitable for the full temperature range of the simulation chamber, and thus a different solution had to be found. Metal C-ring seals are suitable for low and high temperatures and can be used in a pressure range from vacuum to more than 5000 bars. They also need a low to moderate sealing load which implies that they can be used

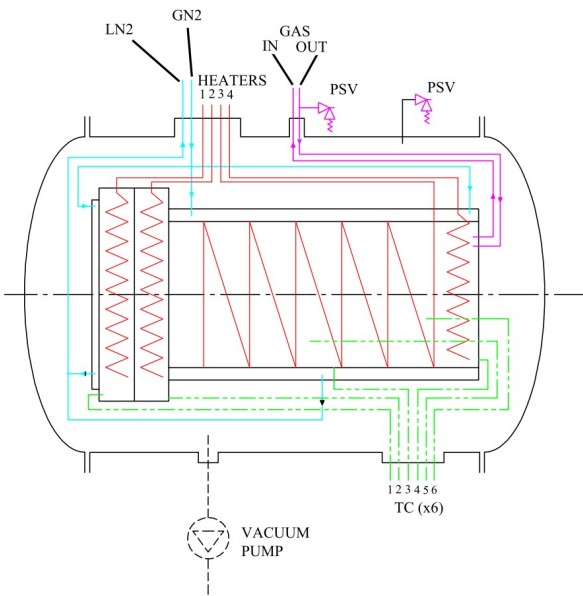

**Figure 2.** Schematic view of the sub-systems. The heaters are indicated in red, the liquid nitrogen circuit in blue (LN2 and GN2 indicate liquid nitrogen input and output), the thermocouples (TC) in green and gas supply lines in violet. PSV indicates the pressure valves on the gas pressure inlet and outer chamber.

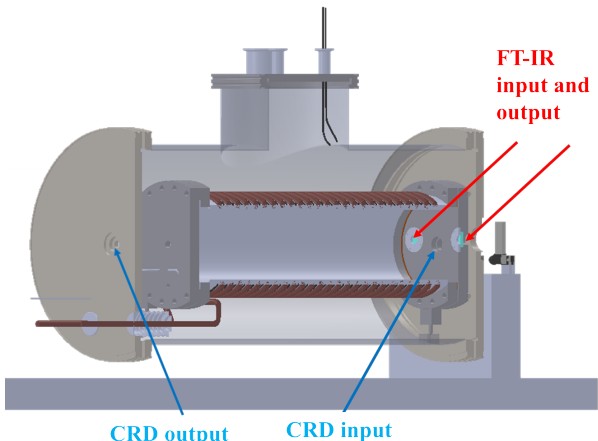

**Figure 3.** Position of the windows dedicated to FT-IR spectroscopy (red arrows) and to CRD experiment (blue arrows)

with rather soft flange materials. This does not include all optical materials however; when we tried to use ZnSe windows, we observed that the C-rings caused an indent in the material after heating. Based on the hardness of the different optical materials 140 (see Table 2), also $BaF_2$, $CaF_2$ and NaCl are probably too soft to be used with C-rings. On the other hand sufficiently hard

materials, such as sapphire, Si, fused silica, fused quartz, BK7 glass and borosilicate glass are compatible with metal C-rings. The minimum thickness of an optical window can be calculated by using the formula

$$t_{min} = K \times R \sqrt{\frac{P \times F}{M_R}} \tag{1}$$

where $t_{min}$ is the minimum thickness, K is the fraction of the window subject to a pressure difference, P is the pressure difference, R is the radius of the window, F is the safety factor and $M_R$ is the modulus of rupture. The commonly used safety factor used for pressure vessels is between 3.5 and 4. The minimum thickness for windows of different optical materials has been calculated for a safety factor of 4, K =0.8, a pressure difference of 60 bars and a radius of 12.5 mm and displayed in Table 2. Since the simulation chamber should support many thermal cycles, it is preferable to use materials with a good resistance to thermal shocks, which relates to a low thermal expansion coefficient and a good thermal conductivity. All materials in Table 2, except for $BaF_2$, $CaF_2$ and NaCl fulfill this requirement. Then of course the optical properties are important, in terms of spectral transmission and overall transmission. The materials with a high index of refraction (Si, ZnSe and ZnS) have a reduced transmission, which may have an important impact on the optical signal at the exit of PASS$x$S, considering that all optical experiments need two window transits. This problem might be mitigated by using optical coatings, but one must verify that the coatings are stable at high temperatures. For what concerns certification of optical windows for high pressures and temperatures, certified sight windows are available in fused silica, fused quartz, sapphire and borosilicate (see e.g. https://rayoteksightwindows.com/). Presently the simulation chamber is equipped with 3 mm thick sapphire windows.

Also the outer chamber has to be equipped with two sets of windows, but since these are at ambient temperature and support a pressure difference of 1 bar, vacuum sealings and window properties do not pose severe requirements. Presently PASS$x$S has been equipped with windows in $CaF_2$ and fused silica for the outer chamber.

## 5  Safety

Operating the simulation chamber in a laboratory requires that several safety measures are in place. The main risk is the rupture of one of the optical windows in the high pressure chamber. A safety valve has been mounted at the gas inlet to avoid that the pressure in the inner chamber exceeds the maximum operating pressure. The cell will not be operated at pressures exceeding 60 bars, although the design maximum pressure is 100 bars. But a rupture of a window might occur and this would cause a flow of the gas in the outer chamber and a rupture of the windows of the outer chamber, with a consequent emission of the gas in the laboratory. A rupture disc, also known as a pressure safety disk has been mounted on the outer chamber to avoid the rupture of the windows in the outer chamber and has been combined with controlled expansion of the escaping gas. One should bear in mind that the volume of the outer chamber is about 5-6 times ( 62 L vs 11 L) larger than the inner chamber, and thus the maximum pressure in the outer chamber will be reduced due to the expansion in the larger volume and can be easily handled after the rupture disc. In case of failure of the temperature control, or if a too high temperature will be reached, a redundant temperature measurement has been put in place to shut down the heating when the temperature reaches a preset threshold.

## 6 Optical Interfaces and alignment

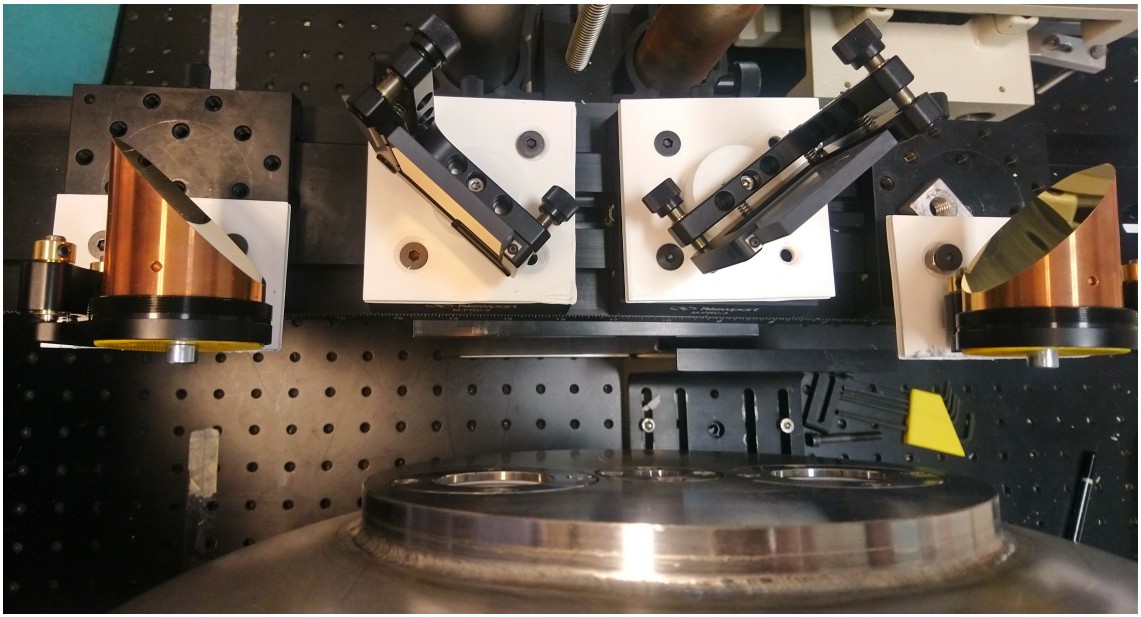

**Figure 4.** The transfer optics for the FTIR is shown.

Figure 4 shows the transfer optics for the FTIR spectrometer. The collimated parallel beam of the FTIR spectrometer is focused with an off-axis parabolic mirror (focal length 30 cm) and deflected by a plane mirror to produce a focal point at the
175 entrance of the multipass White cell optics (CIC Photonics, https://cicp.com) inside the pressure chamber and a spot with a maximum diameter of 25 mm on the two focusing mirrors of the White cell. A similar combination of a flat folding mirror and an off-axis parabolic mirror is producing a collimated exit beam, that is successively focused with short focal length off-axis parabolic mirror on a detector. The White cell optics inside the pressure chamber has a base length of 20 cm, and has been aligned to obtain a pathlength of 3.2 m. An effective optical path length of 9.6 m is feasible, but presently we choose for a more
conservative alignment, to facilitate the coupling with the FTS. The mirrors of the White cell are fabricated out of stainless steel and have protected gold-coated mirrors with proprietary strata that bonds the reflective coatings and protects the mirrors from chemical attack. These mirrors provide over 98.5% reflectivity in the mid-infrared region, thus providing an extraordinary optical power throughput. The FTIR spectrometer, the transfer optics and the simulation chamber have all been mounted on the same anti-vibration optical table and provide a stable alignment. The inner chamber has been mounted on three ceramic
(MACOR) supports, one of these can be seen in Figure 1, fixed at the height of the optical plane of the multipass optics. We estimate that the central plane of the sample chamber will be stable within $\pm$ 0.2 mm with respect to the position at room temperature, during the heating and cooling processes. We calculated, by using the thermal expansion coefficients, that the base length of the White cell optics will vary less than $\pm$ 1 mm during heating and cooling. These small displacements, due to

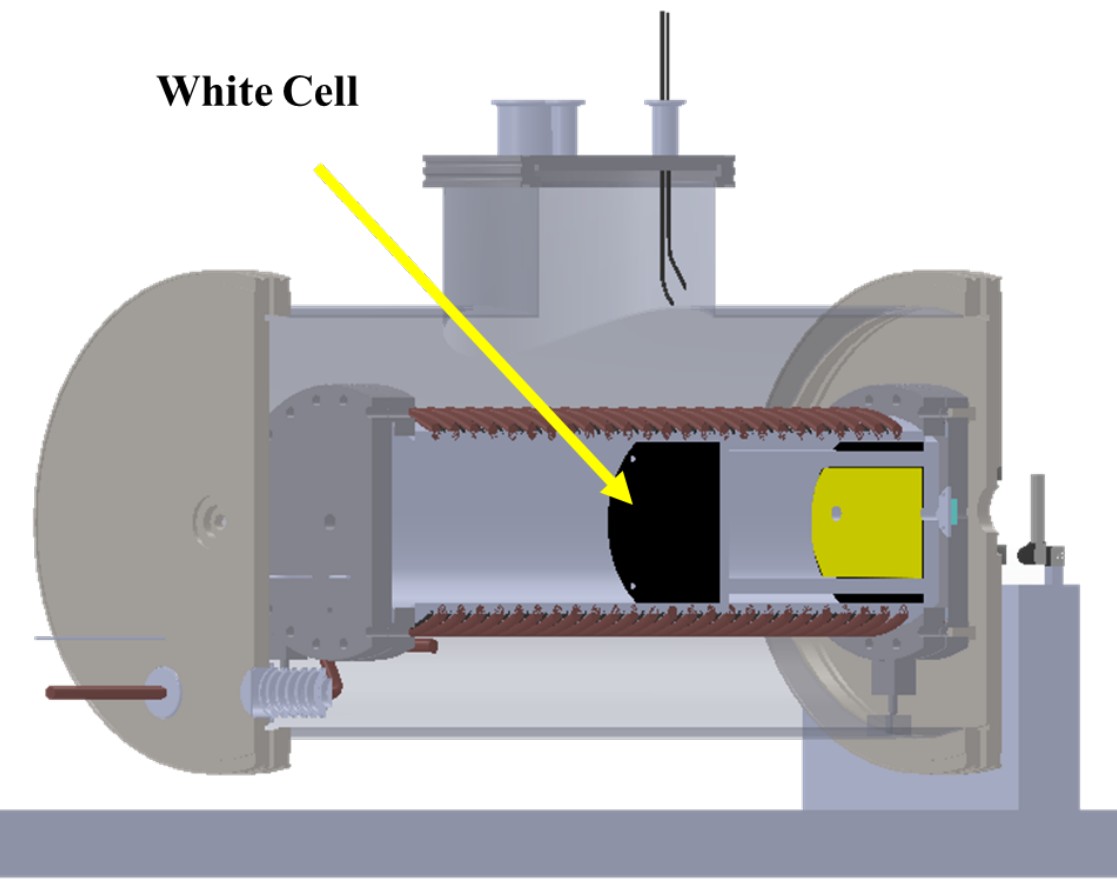

**Figure 5.** The configuration of the White cell installed inside the sample chamber. The total internal length of tha sample cell is about 0.6 m, while the White cell occupies about 0.25 m.

the thermal expansion or contraction, will not affect the alignment of the FTS beam significantly. The optical configuration of
the White cell fixed inside the sample chamber is shown in Figure 5.

## 7   Tests of the PASS$x$S

### 7.1   Heating and cooling tests

The heating is performed by using four tubular resistors, three of them used to heat the flanges and the fourth to heat the body, as shown in Figure 2. The cooling is achieved with a flow of liquid nitrogen which is divided in separate circuits of
tubings which have been dimensioned to obtain a heat exchange proportional to the masses to be cooled. A dedicated PID controller has been developed using a software in Labview to stabilize the temperature of the sample chamber. Figure 6 shows

a typical cooling and heating test. The temperature of various parts of PASS$x$S has been measured with 6 thermocouples. Two thermocouples are suspended in the gas, at different distances from the welded flange and are called long and short. The other four thermocouples have been mounted on 4 parts of the absorption cell; welded flange, optical flange, the cylindrical body of the cell and the counter flange. Each of these parts is heated by a separate heater.

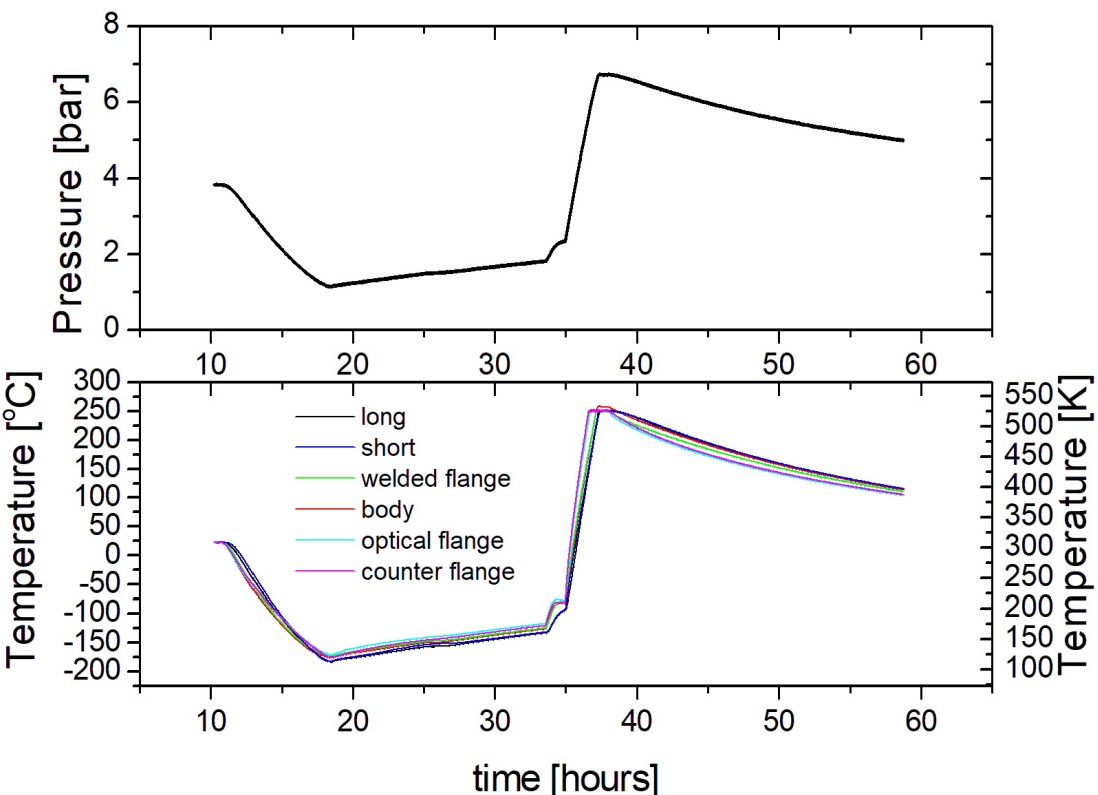

**Figure 6.** Cooling and heating test of PASS$x$S with about 3.7 bar of argon in the sample chamber at room temperature. Upper panel: The pressure of the Ar is indicated in bars. Lower panel: The blue and black curves refer to the temperatures recorded by the short and long thermocouples put inside the inner cell (6 and 5 in Figure 2). The light blue, green, violet and purple curves refer to the temperatures acquired by the other thermocouples (4,3,2,1 in Figure 2) .

In order to measure the maximum heating and cooling rates of the cell, as well as the temperature drift in absence of heating and cooling we have filled PASS$x$S with about 3.75 bar of argon at room temperature and cooled it with a liquid nitrogen flow up to a temperature of 90 K. The maximum cooling rate is about 32 K/ h. At 90 K the liquid nitrogen flow has been stopped, leading to a gradual warming of the sample chamber with a rate of about 3.3 K/h. At about 140 K we observe an onset of heating, but no linear behaviour. This might be due to residual nitrogen gas in the circuit. From 180 K a linear heating

is observed, caused by switching on the four heaters at full power. The sample chamber is then actively heated with the four heaters indicated in Figure 2 from 180 to 540 K, with an average heating rate of 144 K/h. The heaters have been switched off at 540 K and the chamber cooled down with a cooling rate of about 6.2 K/h. This shows that heating is much more efficient than cooling and that in absence of cooling and heating the temperature of the sample is relatively stable and reaches an equilibrium with the ambient temperature with a rate of 3-6 K/h. This slow equilibration is due to the large heat capacity of the sample cell and the good thermal insulation.

If we assume that the gas in the sample cell is in thermal equilibrium, the measured pressure can be used to calculate the temperature of the gas. We use the NIST database for fluids and gases (https://webbook.nist.gov/chemistry/fluid/) to obtain a density of 0.1562 mol/l, given the initial pressure and temperature. Since the heating and cooling processes are isochoric we can obtain the temperature from the pressure measurements by using the NIST tables and compare this with the temperatures measured by the thermocouples in the gas. In general there is a good agreement (better than $\pm$ 2 K) between the temperature obtained from the NIST tables and that measured with the thermocouple in the center of the cell (referred to as "long") if the cooling and heating is slow enough for allowing the gas to reach an equilibrium with the cell (see Figure 7) . If the heating or cooling is too fast, the measured temperatures (long and short) may lag behind with respect to the NIST temperature for different pressures. Probably the temperature obtained from the measured pressure is more representative for the gas temperature than the one obtained from the thermocouple, since it is an averaged quantity.

FTIR measurements at relatively low spectral resolution ( 1-2 cm$^{-1}$ FWHM) require only a few minutes to record 100-200 scans, sufficient for a good signal-to-noise ratio. The temperature of the gas during such a measurement can be considered constant up to a good approximation. The drawback of such a passive thermal stability is that it would take between 40 and 50 hours for the sample to return to ambient temperature, starting from the maximum or minimum temperature. Another approach would be to cool the sample down to 90 K and then heating the cell with moderate power to reach 500 K in 4-5 hours. It is important to verify if the different parts of the sample chamber are in thermal equilibrium during passive cooling and heating. We observed that during the cooling process and the successive passive heating the temperature differences between the different parts of the cell as measured by the four thermocouples, were small (typically less than 10 K).

However, we observed that when we supplied the maximum power to all four heaters, some parts of the cell (the optical flange and its counter flange) were much hotter than the body and the welded flange, so we regulated the power to each heater in a way that all parts would have similar temperatures at all times. This resulted in much smaller temperature differences during the active heating process (typically less than 10 K).

The temperature of the cell has been also actively stabilized by using a PID (proportional-integral-derivative) feedback on the heaters. When the set point has been reached the power supply for each heater is controlled by a separate PID control which regulates the supplied power and switches the heaters on and off. This method is used for temperatures above ambient temperature. Figure 8 shows how the cell is heated in three steps and stabilized in temperature at 50, 100 and 150 °C. It can be observed that the temperatures of the different parts of the cell are in excellent equilibrium (within a few degrees) once the PID regulation takes effect, while the gas temperature is reaching an even more stable temperature ($\pm$0.5 K or better) after 15 to 20 minutes. Active temperature stabilization below room temperature requires that the liquid nitrogen flow is always

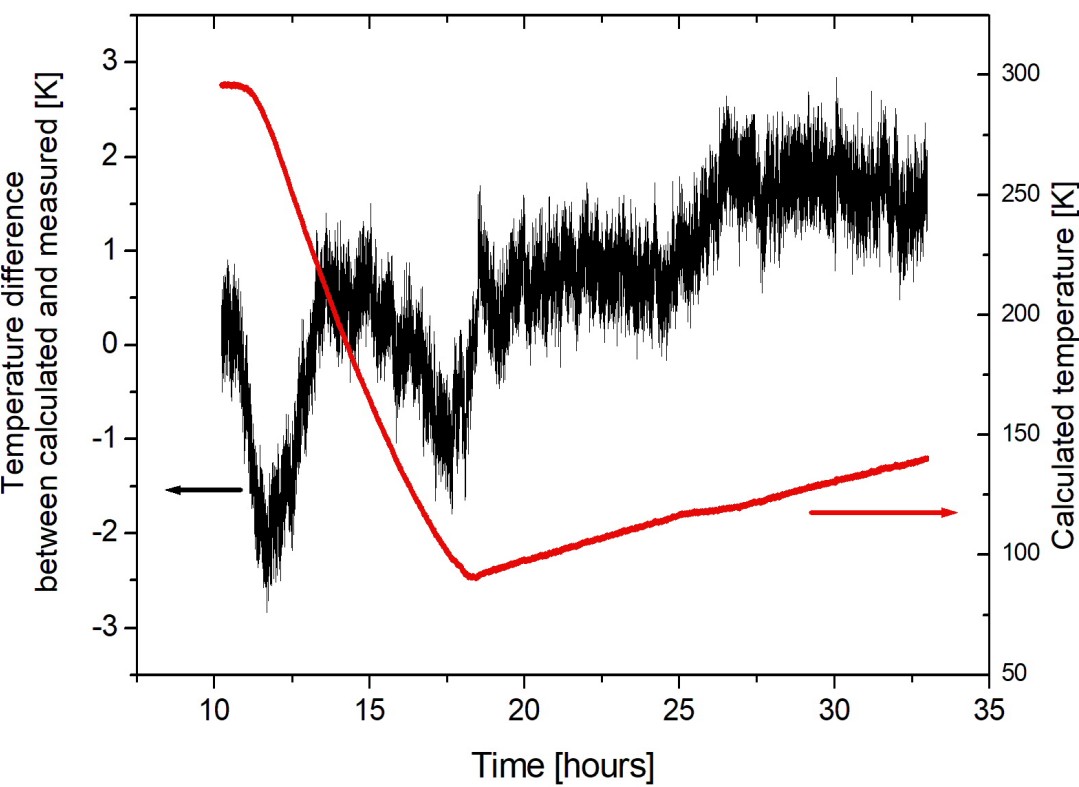

**Figure 7.** The difference of the temperature obtained from the NIST tables for the pressure of the Ar gas with respect to the gas temperature measured by the "long" thermocouple has been displayed (black curve). The agreement is $\pm$ 2 K. The red curve corresponds to the gas temperature obtained from the NIST tables.

active to counteract the heating regulated by PID control. During cooling the liquid nitrogen flow is maximum until the preset temperature is reached and is reduced to maintain roughly the set point. Fine tuning is obtained by regulating the four heaters by PID as explained before.

The difference of the temperature measured by the two thermocouples mounted inside the absorption cell is a measure for the homogeneity of the gas temperature in the cell. We observed that this difference was always less than $\pm 2.5$ K for the full temperature range from 90 K to 525 K, being slightly better at the high end of the range (better than $\pm 1.8$ K), when heating and cooling were turned of (the second and final part of the curves in figure 6), while during the heating and cooling processes the differences were oberved to be as large as 20 K in the worst case. When the cell was actively stabilized in temperature the difference between the temperatures measured by the two thermocouples in the gas was always less than $\pm 0.7$ K when the cell was above ambient temperature and less than $\pm 1.5$ K below ambient temperature. Apparently the simultaneous flow of liquid

nitrogen and the heating supplied by the four PID regulated heaters creates some extra non-homogeneity of the temperature in the gas.

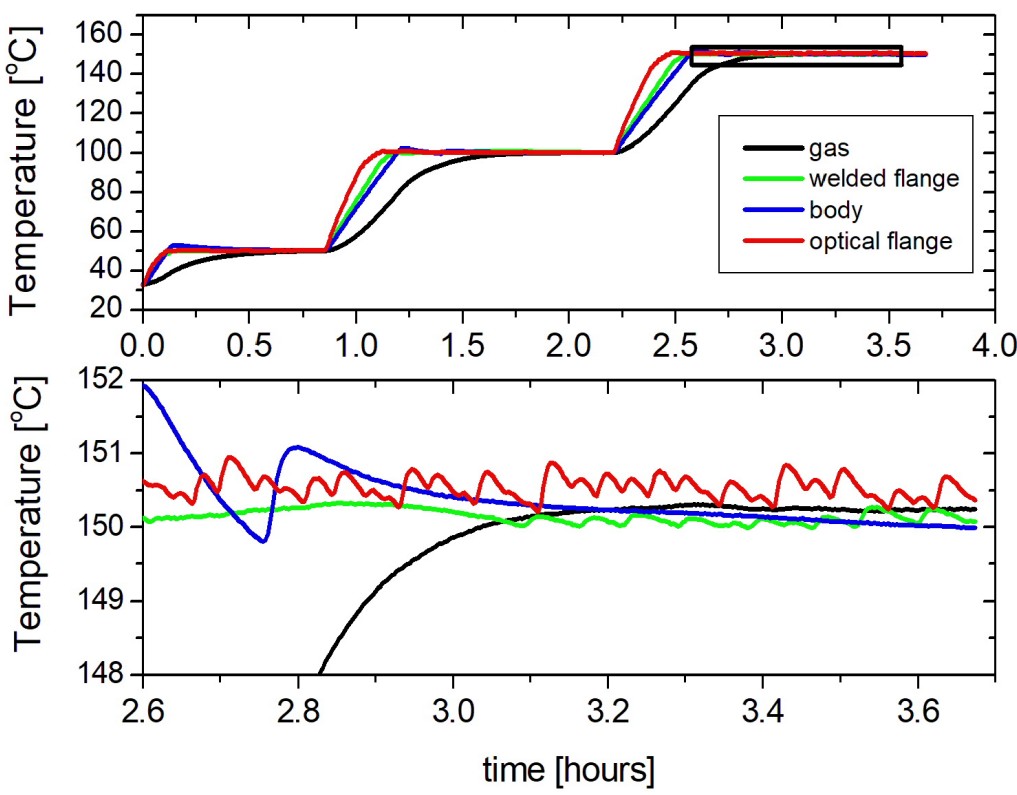

**Figure 8.** Upper panel: heating and temperature stablization in three steps. Lower panel (enlarged part of the upper panel, indicated as a black box in the upper panel) : Temperature stability of the different parts of the cell and the gas during active stabilization.

## 7.2 Tests of the FTS coupled to the multipass cell

The FTS ( a Bruker model Vertex 80) was used to acquire spectra by averaging 256 scans with a resolution of 2 cm$^{-1}$. The detector employed was a narrow band liquid nitrogen cooled MCT detector with a ZnSe window, with a specific sensitivity D* of $2 \times 10^{10}$cmHz$^{1/2}$W$^{-1}$ from 600 to 6000 cm$^{-1}$. The standard deviation of the absorbance measured with this set up was estimated to be $3 \cdot 10^{-5}$ from the signal of an empty cell. The multipass cell mounted inside PASS$x$S is an open commercial White cell, with a base length of 20 cm. The effective optical path can be varied from a minimum of 0.8 m to a maximum of 9.6 m in steps of 80 cm.. Here we have aligned the cell with 16 passes before inserting it in PASS$x$S, corresponding to a 3.2 m pathlength. It should be noted that the coupling of the FTS to the White cell is not straightforward, since the propagation of

the focused light beam can not be observed once inside the two vacuum chambers. After an initial alignment with the visible light emitted by a HeNe laser, the transfer optics have been fine-tuned to obtain a maximum output at the exit window.

The effective optical path length of the White cell inside the sample chamber (see Figure 5) has been verified by performing a measurement of a known quantity of carbon dioxide and comparison with a spectral simulation. To this purpose the sample cell has been filled with carbon dioxide at a pressure of 4.3 bar at room temperature (294 K). For this measurement we used a MCT detector, a KBr beam splitter and a resolution of 2 $cm^{-1}$. Using a free on line tool for the simulation (https://spectra.iao.ru/en/en/mixt/spectr/) a synthetic spectrum was obtained. The results are shown in Figure 9, where the red curve refers to the measured spectrum while the black curve represents the simulation with an optical pathlength of 3.2 m. The excellent agreement between the two curves confirms without any doubt that the optical pathlength is 3.2 m, considering that the pathlength varies in steps of 80 cm ( 4 passes).

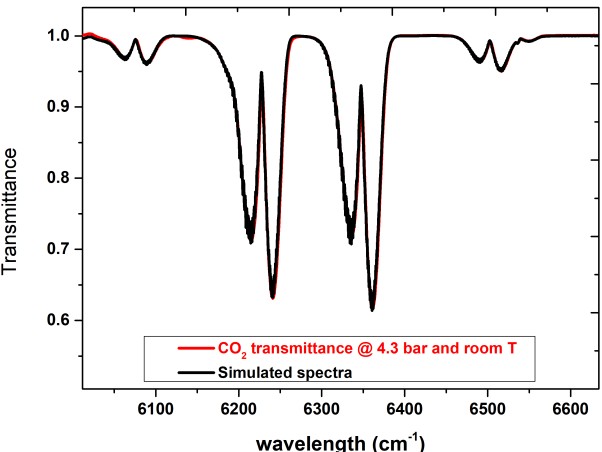

**Figure 9.** Comparison between acquired $CO_2$ spectra @ room temperature and 4.3 bar (red curve) and simulated (black curve)

### 7.3  Comparison of PASS$x$S spectra with previous measurements.

One of the principal goals of PASS$x$S is to measure collision induced absorption (CIA) bands important for planetary atmospheres. CIA bands are spectral features generated by inelastic collisions and are thus important for dense environments such as the atmospheres of gas planets. While the intensity of allowed infrared absorption bands is proportional to the molecular density, the intensity of CIA bands is proportional to the square of the density. While the absorption cross section can be expressed as the absorption coefficient in $cm^{-1}$ divided by the molecular density in $mol \cdot cm^{-3}$, the CIA cross section can be expressed in $cm^5 mol^{-2}$ or in $cm^{-1} amagat^{-2}$, where 1 amagat is 44.615 $mol \cdot m^{-3}$.

The error in these parameters depends mainly on the accuracy of the absorbance measured. The main error in the determination of the absorbance is the baseline error. Ideally the absorbance should be zero in absence of absorption, but in practice

the baseline after recording and processing an actual absorption spectrum and a background spectrum is not exactly zero for the full spectral range, but might have a small offset or a wavenumber dependent offset, being typically less than 0.01. Often a baseline correction is applied, adding an offset to obtain zero absorption. After the baseline correction, a residual baseline error might be in the order of 0.0001 to 0.001. The second error source in the absorbance is due to the detector noise, thermal noise and ADC (analog-to-digital converter) noise. Part of this noise can be mitigated by averaging over many FTS scans. We

have observed noise levels of the order of $10^{-5}$ with 256 averages.

In order to demonstrate how PASS$x$S improves upon previous measurements, we recorded a spectrum of carbon dioxide at 243 and 323 K at a density of 16.03 amagat (see Fig. 10). Note that for this density carbon dioxide starts to liquify at 238.4 K. The spectrum shows three bands belonging to the Fermi triad ($2\nu_1$,$\nu_1+2\nu_2$,$4\nu_2$) at 2544, 2670 and 2796 cm$^{-1}$, and the $\nu_2+\nu_3$ combination band at 3015 cm$^{-1}$. All these bands are CIA bands and have been studied previously (Stefani et al., 2018;

Baranov et al., 2003).

Stefani et al. (2018) recorded absorption spectra of carbon dioxide in a 2 cm long heatable cell to study CIA bands in the range from 2500 to 3500 cm$^{-1}$, with ultra grade high purity carbon dioxide (99.995 %) for densities from about 15 to 40 amagat and temperatures between 300 and 475 K. Baranov et al. (2003) reported room temperature CIA bands at various densities as well as low temperature spectra at 1 atmosphere with an optical path of 30.09 m. In both cases, the strong contributions of the $\nu_3$

fundamental band at 2350 cm$^{-1}$ and a weaker Fermi doublet ($2\nu_1$,$\nu_1+2\nu_2$,$4\nu_2$) around 3700 cm$^{-1}$ had to be subtracted, as well as contributions of the $^{16}O^{12}C^{18}O$ isotopomer, before the CIA bands could be studied in detail. Here we compare the spectra recorded with PASS$x$S with those reported previously. First of all, PASS$x$S allows to record spectra at temperatures below and above room temperature, with the same experimental conditions, which makes a study of temperature dependence easier than by comparing measurements taken by different instruments. We assume that the accuracy of the measured absorbance , due

to baseline errors and detector related noise, is similar in all experiments, and calculated the observed absorbances in all three experiments, in order to compare signal-to-noise ratios. In Stefani et al. (2018) the binary integrated absorption coefficient has been reported, and Baranov et al. (2003) report the integrated intensity (note that in Table 1 the units should have been cm$^{-2}$amagat$^{-2}$), although the term integrated CIA absorption cross section would have been more correct in both cases. When we take into account the optical absorption length and the densities we can calculate the integrated absorbance measured in

each experiment (see Table 3) for the 2670 cm$^{-1}$ band. The average absorbance can be obtained by dividing the integrated absorbance by the length of the integration interval (2590 - 2730 cm$^{-1}$).

Note that the present data show a higher average absorbance for the 2670 cm$^{-1}$ band at both temperatures, having a longer optical path than the short heated cell and a higher density than the cooled cell. Note that the weak band at 2544 cm$^{-1}$ has been observed for the first time with a good signal-to-noise ratio.

We are now able to investigate in detail the temperature dependence of the three CIA bands of the Fermi triad as well as the $\nu_2+\nu_3$ absorption band. One can observe that all three bands of the Fermi triad have a strong temperature dependence, while the band at 3015 cm$^{-1}$ has not. This is in good agreement with previous literature data (Stefani et al., 2018).

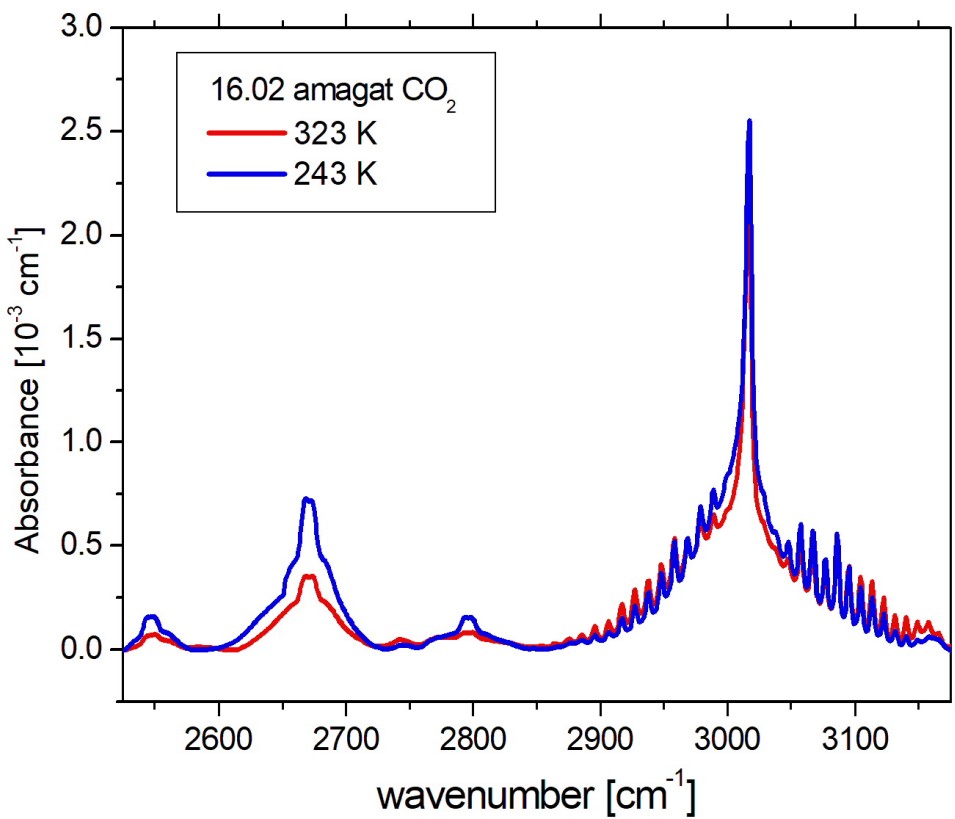

**Figure 10.** The absorption spectra of 16.03 amagat of carbon dioxide at 243 and 323 K, recorded with PASS$x$S, after subtraction of the strong contributions of the $\nu_3$ fundamental band at 2350 cm$^{-1}$ and a weaker Fermi doublet ($2\nu_1$,$\nu_1$+$2\nu_2$,$4\nu_2$) around 3700 cm$^{-1}$, as well as contributions of the $^{16}O^{12}C^{18}O$ isotopomer

## 8    Conclusions

A novel absorption cell for spectroscopy of planetary atmospheres has been designed, constructed and tested. The cell has shown to have an excellent passive thermal stability, remaining within $5\,^\circ$C of the set temperature in an hour of time. Active temperature stabilization has also been demonstrated and a temperature stability of $\pm$ 1 degree, or better, on the time scale of one hour has been obtained.

A maximum heating rate of about 145 K/h has been observed while the cooling rate is about 30 K/h. A full temperature run from the lowest to the highest temperature in one day would require an overnight cooling to the lowest temperature, followed by a heating with a heating rate of about 60 K/h. A data acquistion of the Fourier Transform Spectrometer recording 256 scans at medium resolution, would allow to obtain a full spectrum at regular intervals of a few degrees. A comparison with previous

measurements of CIA bands of carbon dioxide showed that PASS$x$S has superior sensitivity with respect to previous studies and allows to measures cold and hot spectra in the same measurement session.

We are planning to mount a resonant CRD cavity in PASS$x$S, in order to perform more sensitive absorption measurements for selected wavelengths.

*Author contributions.* GP and MS developed the concept of PASS$x$S. AB was responsible for the 3D design of the instrument. SS, GP, AB and DB took care of assembling, tests and PID development and implementation. SS performed all test measurements. MS prepared the manuscript with contributions from all co-authors.

*Competing interests.* The authors declare that they have no conflict of interest.

*Acknowledgements.* The authors acknowledge funding by ASI-INAF under the contract n. 2018-25-HH.0. The scientific activities have been supported by JUICE and Progetto Premiale INAF WOW 2013.

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

| type of cell | pressure | temperature range | P.L [m] | window | Reference |
| --- | --- | --- | --- | --- | --- |
| cell in furnace | <4 atm | 300 K < T < 1200 K | 0.2 | $Al_2O_3$ | Abu-Romia and Tien (1966) |
| cell in furnace | <60 bar | 291 K < T <751 K | 0.044 | $Al_2O_3$ | Hartmann and Perrin (1989) |
| cell in furnace | < 50 atm | 300 K < T < 900 K | 0.37 | $Al_2O_3$ | Rieker et al. (2007) |
| cell in furnace | < 200 bar | 300 K < T < 650 K | 0.02 | ZnS | Stefani et al. (2013) |
| cell in furnace | < 100 bar | 300 K < T < 1000 K | 0.03 | $Al_2O_3$ | Christiansen et al. (2016) |
| cell in furnace | < 30 atm | 300 K < T < 800 K | 0.21 | $CaF_2$ | Almodovar et al. (2019) |
| cell in furnace | < 100 atm | 300 K< T < 1200 K | 0.37 | $CaF_2/Al_2O_3$ | Schwarm et al. (2019) |
| in vacuum cell | 10 bar | 87 K < T < 120 K | 165 | not specified | McKellar et al. (1970) |
| in vacuum cell | 1 - 5 bar | 190 K< T < 300 K | 512 | $CaF_2$ | Ballard et al. (1994) |
| in vacuum cell | < 1 bar | 120 K< T < 300 K | 2540 | not specified | Horn and Pimentel (1971) |
| in vacuum cell | < 1bar | 100 K< T < 300 K | 500 | not specified | Briesmeister et al. (1978) |
| in vacuum cell | < 1 bar | 160 K< T < 300 K | 40-1500 | not specified | Kim et al. (1978) |
| in vacuum cell | < 1 bar | 20 K< T < 296 K | 12.49 | $CaF_2$ | Mondelain et al. (2007); Guinet et al. (2010) |
| in vacuum cell | < 10 atm | 215-470 K | 96 | $BaF_2/KBr$ | Shetter et al. (1987) |
| polystyrene | < 1 bar | 123 K < T < 423 K | 3 | $CaF_2$ | Schermaul et al. (1996) |

**Table 1.** The Table lists some relevant properties of a number of coolable and heatable absorption cells reported in literature. The Table is not an exhaustive list of such cells, but is intended to give an example of what has been achieved in the past.

| material | spectral transmission (nm) | index of refraction | maximum transmission | thickness (mm) | Hardness Knoop | thermal expansion coefficient ($10^{-6}$K$^{-1}$) | thermal conductivity (Js$^{-1}$m$^{-1}$K$^{-1}$) |
|---|---|---|---|---|---|---|---|
| BaF$_2$ | 150-12000 | 1.48 | 0.93 | 9.51 | 82 | 18.1 | 11.7 |
| CaF$_2$ | 130-10000 | 1.43 | 0.94 | 8.16 | 178 | 18.8 | 9.7 |
| NaCl | 250-16000 | 1.54 | 0.91 | 31.75 | 18 | 40 | 6.5 |
| ZnSe | 500-16000 | 2.4 | 0.66 | 6.64 | 120 | 7.1 | 18 |
| ZnS | 370-13500 | 2.37 | 0.67 | 5.94 | 210 | 6.6 | 16.7 |
| Sapphire | 200-5000 | 1.77 | 0.85 | 2.33 | 2000 | 4.5 | 34.6 |
| Si | 1500-8000 | 3.46 | 0.39 | 4.41 | 1150 | 4.15 | 163.3 |
| Fused silica | 150-3200 | 1.46 | 0.93 | 7.1 | 600 | 0.55 | 1.5 |
| Fused quartz | 260-2500 | 1.46 | 0.93 | 7.1 | 600 | 0.55 | 1.3 |
| BK7 | 350-2500 | 1.52 | 0.91 | 12.12 | 480 | 7.1 | 1.1 |
| Borosilicate | 350-2800 | 1.51 | 0.92 | 6.86 | 470 | 3.3 | 1.14 |

**Table 2.** The Table lists the most important optical, thermal and mechanical properties of a variety of optical window materials. The values reported should be taken as indicative values, since different values can be found in literature and in specifications of commercial optical materials.

| temperature [K] | density [amagat] | optical path [cm] | int. absorbance [$cm^{-1}$] | average abs | ref |
|---|---|---|---|---|---|
| 235 | 1.12 | 3009 | 0.59 | 0.0042 | Baranov et al. (2003) |
| 243 | 16.03 | 320 | 9.05 | 0.0646 | this work |
| 322.8 | 40.2 | 2 | 0.26 | 0.0019 | Stefani et al. (2018) |
| 323 | 16.03 | 320 | 3.86 | 0.0276 | this work |

**Table 3.** The Table displays the experimental conditions for the CIA band at 2670 $cm^{-1}$ and the calculated integrated absorbance and average absorbance.