# Peer review of "A simulation chamber for absorption spectroscopy in planetary atmospheres."

_Atmospheric Measurement Techniques, 2021_

## Referee Comment (RC1)

https://doi.org/10.5194/amt-2021-245

*A simulation chamber for absorption spectroscopy in planetary atmospheres* by Marcel Snels, Stefania Stefani, Angelo Boccaccini, David Biondi, and Giuseppe Piccioni submitted to AMT.

**Summary**

This paper report the design and test of a simulation chamber including a White cell (and in the future a high finesse cell for CRDS measurements) allowing recording transmission spectra of different gas mixtures of atmospheric interest over large temperature (from 100 K to 550 K) and pressure (up to 60 bar) ranges.

**General comments**

This paper is well-written and the design of the simulation chamber is well-explained and described. Performances in terms of temperature stability and homogeneity are particularly good over the whole temperature range and sufficient for most of the applications dedicated to the planetary atmospheres if we except the Earth's atmosphere where, for example, green house gases monitoring requires a very accurate knowledge of the temperature dependence of the spectroscopic parameters and thus a better temperature homogeneity. My fist remark concerns the discussion about the recorded spectra which needs to be fleshed out: nothing is said about the noise level and the base line stability of the spectra while these aspects are essential to obtain accurate spectroscopic parameters and cross-sections for the continua. The large volume of the cell could also be a problem when using expansive isotopically-enriched species or hazardous gases. This has to be mentioned. A table summarizing the gas temperature homogeneity and stability at the different achievable temperatures has to be included in the paper.

This paper deserves to be published in AMT with minor modifications.

**Minor remarks/comments**

L27: …FTS mea**s**urements.

L67-71: The following reference should be cited: Guinet, M., et al. *Performance of a 12.49 Meter Folded Path Copper Herriott Cell Designed for Temperatures Between 296 and 20 K*, Applied Physics B 100 (2010): 279–282.

L97: *double wall stainless steel tubing*: this is not clear to me what it exactly means. Could the authors reformulate this sentence?

Fig. 2: Indicate in the caption what TC, PSV, LN2, GN2 mean. It will be more comfortable for the reader.

L171: Please give information about the detector and its electronics (type of the detector, detectivity, bandwidth…).

L181: *The base length of the White cell optics will vary less than +/- 1 mm...* How this value is determined? Measured of calculated from the thermal expansion coefficient?

Fig. 5: In the caption please recall the total length of the sample cell and of the White cell.

Fig. 6: The legend on the figure doesn't correspond to what is written in the caption.

L207: *calculated temperature*: how is this temperature calculated?

Fig. 7: Why not making the comparison by doing temperature steps and waiting for temperature to stabilize? What are the temperatures measured by the other thermocouples? The accuracy of the TC has to be given.

In the caption of this figure: *The red curve corresponds with the calculated gas temperature* has to be replaced by: *The red curve corresponds **to** the calculated gas temperature.*

Fig. 8: The area zoomed on the lower panel has to be indicated (by a rectangle for example) on the upper panel.

In the caption: *stabliaztion -> stabilization*.

L230: *stabile -> stable*

L230-235: The authors have to clearly distinguish between the gas temperature stability with time and the gas temperature homogeneity (in space). Both values have to be given in a table for different temperatures over the entire temperature range (see my general comment).

L237: Please indicate the reference of the commercial White cell.

L239: *corresponding with -> corresponding **to***

L244-245: *carbondioxide -> carbon dioxide*

L243-249: It is important to recall here that the n-1 path length is 2.4 m and the n+1 path length is 4 m so that there is no doubt about the determined path length even if there are some uncertainties on the intensities of the $CO_2$ transitions.

Fig. 10: Add also a fit of the spectra and show the fit residuals on a distinct panel. This will allow discussing the noise level of the spectra.

In the caption: *carbondioxide -> carbon dioxide*

---

## Author Response (AR1)

Answers to the referees:

We thank both referees for their comments and suggestions. The main point regards a meaningful comparison with previous studies and a discussion of the improvement with respect to the state of the art.

An example of new measurements with PASSxS has been discussed in detail and compared with previous measurements at temperatures below and above room temperature. Since one of the principal goals of PASSxS is the study of collision induced spectra at a range of temperatures, we've chosen to remeasure several collision induced bands of carbon dioxide at different temperatures at a pressure of about 16 bar.  At this density carbon dioxide starts to liquify around 239 K, which puts a limit to the lowest temperature that could be obtained without changing the gas density. It is also well known that the CIA absorption coefficients of carbon dioxide tend to decrease with higher temperatures, which renders the weaker bands difficult to observe for high temperatures.

The significance of the data extracted from these spectra (band integrated CIA absorption cross sections) depends strongly on the signal to noise ratio, which has been calculated here for the absorbance observed in all studies (the SNR of the absorption cross sections equals the SNR of the absorbance). We assume that the detector noise (which is mitigated by averaging the spectra) and baseline errors are similar for all studies, since no specific information is available in [Baranov2003]. The results have been displayed in a new Table 3. The result is that PASSxS produces more accurate parameters for both temperatures.

Also the temperature stability has been discussed in more detail. First we consider passive cooling and heating, when the cell is free to drift towards the ambient temperature,  and we find a drift of 3-6 K/h. In this regime the temperature homogeneity, as inferred from the difference of the temperature measured by the two thermocouples in the gas at different positions (long and short) is always better than ±2.5 K.

When an active temperature stabilization is applied,  by PID regulation of the heaters, the temperature homogeneity is better, about ±0.7 K when the cell temperature is above room temperature and about ±1.5 K when it is below.

For what concerns the accuracy of the absorbance measurements we have discussed the possible error sources in more detail. The main error in the determination of the absorbance is the baseline error. Ideally the absorbance should be zero in absence of absorption, but in practice the baseline after recording and processing an actual absorption spectrum and a background spectrum is not exactly zero for the full spectral range, but might have a small offset or a wavenumber dependent offset, typically this is less than 0.01. Often a baseline correction is applied, adding an offset to obtain zero absorption. After the baseline correction, a residual baseline error might be in the order of 0.0001 to 0.001. The second error source in the absorbance is due to the detector noise, thermal noise and ADC (analog-to-digital converter) noise. Part of this noise can be mitigated by averaging over many FTS scans. We have observed noise levels of the order of $10^{-5}$ with 256 averages.

For what concerns the minor remarks/comments of referee 1, we have corrected all typos and answered as follows:

L67-71. The reference Guinet et al has been added in the text and in Table 1.

L97. A phrase has been added to explain better the term double wall tubing. The inner walls are in contact with the liquid nitrogen and are welded on one side to a larger tube which provides the vacuum, thus avoiding exposure of the cold tube to the environment.

Fig2. The caption has been modified as requested

L171: The information about the detector has been added. The detector employed was a narrow band liquid nitrogen cooled MCT detector with a ZnSe window, with a specific sensitivity D* of $2 \times 10^{10}$ cmHz $^{½}$ W$^{-1}$ from 600 to 6000 cm$^{-1}$.

L181. The base length variation has been calculated from the thermal expansion coefficient of the material. This has been added in the manuscript

Fig 5. The total length of the sample cell and the White cell has been added in the caption

Fig 6. The caption has been corrected

L207. We explained more clearly how we used the tables in the NIST webbook. In practice we use the NIST Tables to find the density corresponding with the initial temperature and pressure. Then we use the Tables for isochoric processes to find the temperature for each measured pressure, given the density, which remains constant.

Fig7. Figures 6 and 7 display how PASSxS behaves when heated and cooled and when left by itself to drift towards the room temperature. This test has the goal to determine heating and cooling rates and to test the thermal stability once heating and cooling are switched off. This has been stated more clearly in the text. Figure 8 shows what happens when we actively stabilize the temperatures with a PID control of the heaters, just as the referee suggests. The accuracy of the thermo couples has been stated in the text. (The accuracy of the type T thermocouples is ±1ºC or ±0.75%, whichever is greater.

The caption of figure 7 has been corrected

Fig 8. A box has been inserted in the upper panel to show which part we display in the lower panel. The caption has been corrected.

L230-235. We discussed more clearly both temperature stability and homogeneity. The homogeneity is approximated by comparing the temperatures measured with the two thermocouples suspended in the gas at different positions (short and long, where short is close to the wall and long is at the centre of the cell)

L237 the reference has been added.

L243-249. We thank the referee for this useful suggestion, which has been added

Fig 10 has been eliminated and a new figure has been inserted and extensively discussed (see also above)

List of modifications of the manuscript:

Some sentences have been rephrased to enhance readability and some typos have been corrected in the manuscript.

The use of the NIST database to obtain temperature from measured pressure at constant volume has been explained.

The temperature stability and homogeneity has been discussed in more detail, both for passive conditions (heaters and cooling off) and with PID stabilization of the temperature. Numbers have been given, derived from measurements as displayed in figures 6,7 and 8. Please note that figure 8 is just an example. This kind of test has been performed for the full temperature range.

The comparison of PASSxS with previous measurements has been rewritten completely, with a new figure 10. A major application of PASSxS is the study of CIA bands, and thus we have remeasured 4 CIA bands of carbon dioxide that have been already measured in the past both at temperatures above and below room temperatures. The comparison is illustrated for two temperatures close to those reported in literature. Assuming similar accuracies for the absorbance, we've calculated the absorbance obtained in all experiments and made a comparison of the absorbances.